# To Laparoscopically Preserve Fertility in Intraabdominal Giant Myoma with Application of Contained In-Bag Morcellation: Mission Impossible?

**DOI:** 10.3390/jcm11154531

**Published:** 2022-08-03

**Authors:** Rajesh Devassy, Luz Angela Torres-de la Roche, Johannes San Juan, Harald Krentel, Sven Becker, Rudy Leon De Wilde, Amr Soliman

**Affiliations:** 1University Hospital for Gynecology, Pius Hospital, University Medicine Oldenburg, Carl von Ossietzky University, 26121 Oldenburg, Germany; rajeshdevassy@gmail.com (R.D.); luz.angela.torres-de.la.roche@uol.de (L.A.T.-d.l.R.); johannes.sanjuan07@gmail.com (J.S.J.); krentel@cegpa.org (H.K.); gyn-sekretariat@pius-hospital.de (A.S.); 2Gynecologic Oncology and Gynecologic Specialties, University Hospital Frankfurt, 60590 Frankfurt am Main, Germany; sven.becker@kgu.de

**Keywords:** myoma, leyomiosarcoma, myomectomy, in-bag morcellation, laparoscopy, video article

## Abstract

A technical video was produced to demonstrate in step-by-step fashion a multiple contained myomectomy of a 20 × 30 cm giant myoma and seven additional fibroids found in the same patient, which required two different types of specimen retrieval bags for the electronic power morcellation. This complete surgical procedure included leiomyomata enucleation, contained in-bag electronic power morcellation, uterine reconstruction and the application of an adhesion prophylactic medical product.

## 1. Introduction

Tissue dissemination can occur during the process of morcellation. The last decade has seen many controversies concerning the method of morcellation, to answer the question whether the dissemination carries any significance or, on the other hand, if it is safe to use an in-bag containment system for morcellation. Handling larger size of myoma specimen with a bag may be challenging to many surgeons. Therefore, this video article’s objective is to demonstrate a step-by-step approach to contained multiple leiomyoma extirpation by in-bag laparoscopic electronic power morcellation. This is the largest organ-saving and fertility-sparing minimally invasive myomectomy, based on total weight, published in the world literature.

The main concern of morcellation is related with the risk of sarcoma and postoperative parasitic leyomioma secondary to tissue dissemination into the abdominal cavity during the procedure [1,2,3,4]. To avoid this, containment systems have been developed. Initial experience of this technique was described since 2014 and was also shown useful in terms of avoiding spillage of tissue or cells during extraction [5]. Especially in cases such as myomectomy and supracervical hysterectomy, the only way to extract the specimen, especially when it is larger, is dividing it into smaller pieces. In-bag morcellation has shown to be feasible [6]. Unsuspected leiomyosarcoma removal with the in-bag morcellation technique has been documented with good outcome [7]. The FDA-2020 final guidelines state that, when morcellation is done appropriately, the contained technique may be adequate and allowed to be performed [8]. Parasitic leiomyomas are a problematic finding after laparoscopic electronic power morcellation when containment is not practiced [9]. Data also refer to patients presenting with adenomyosis masses post hysterectomy by uncontained morcellation [10]. A meta-analysis in the Cochrane database found no significant differences in postoperative outcomes between uncontained and contained morcellation [11].

Guidelines and recommendations, suggested by the FDA, line out that a morcellator can only be used after careful patient selection [12]. Therefore, we are performing ultrasound and color doppler velocimetry for the large fibroids followed by MRI and, if indicated, along with serum markers CA125 to evaluate clinical suspicion of malignancy. Endometrial biopsy is reserved for cases with abnormally thickened endometrium.

Although some studies showed an increase in surgical time due to the morcellation within a containment system, this is only the case during the early learning curve [13]; improvement in necessary operation time with increasing volume of cases and experience with the contained morcellation, even when compared with different types of containment, was published [14]. Complications associated in long-term patient outcome from tissue spilling and dissemination are avoidable even if the containment system has an increased operating time in the beginning of the implementation in the surgical work [15]. In leiomyoma, being the second most common disease for surgery to be performed in women, the possibility of adhesions is high; therefore, the importance of concomitant adhesion prophylaxis is important in all myomectomies [16,17]. Although laparoscopic surgery, being the gold standard in myomectomy, has diminished the incidence of adhesions, it has been demonstrated that the use of an adhesion barrier decreases this possibility significantly [18,19]. Here we present the step-by-step in-bag contained morcellation performed on a patient who presented with a giant myomatous uterus.

## 2. Materials and Methods

We produced a technical video demonstrating in step-by-step fashion a case of laparoscopic multiple myomectomy, utilizing contained in-bag electronic power morcellation. The MRI assessment described a 36-week gravid uteri of clinical size. The entire surgery included multiple leiomyomata enucleation, contained in-bag electronic power morcellation, uterine reconstruction and the application of an adhesion prophylactic medical product. The largest myoma was purely subserous (FIGO type 6), others were intramural myomas (FIGO types 2, 3 and 5). Vasopressin was applied pericapsular as an infiltration medium (0.3 dilution) for bleeding control. The uterine artery and the internal iliac artery were not clipped as they were not accessible due to the large size of the myoma. The uterine cavity was not entered. Myometrium layers were closed with 1-0 VLOC and serosa with 2-0 Vicryl and Monocryl. The blood loss intraoperatively was 1000 mL. The patient received three units of compatible red blood cells, as she was already anemic before the surgery. After morcellation, 1500 mL of bloody fluid was suctioned from the bags.

As represented in the Figure 1, the camera port is inserted through the second sleeve situated on the end of the bag, opposite to the wider bag opening that corresponds to the morcellator port. It avoids air and potential tissue escapes and allows a direct visualization and control of the morcellation procedure.

Because of the presence of a giant fibroid (20 × 30 cm), it was necessary to use two different specimen retrieval bags; Ecosac bag (Espiner Medical—Fannin U.K. Ltd., Stroud, UK) was used to retrieve the giant fibroid and the MorSafe^®^ bag (Veol Medical Technologies Pvt. Ltd., Mumbai, India) was used for the seven remaining tumors. These bags do not tear easily after inadvertent grasping with a non-sharp surgical tenaculum, as expressed by the manufacturers and as we observed throughout our own experience. Figure 2 shows the main steps of the in-bag morcellation process using the MorSafe^®^ system, including a picture of the trocar placement that is recommended when the uteri are grown high up cranially in the abdomen (Figure 2A).

## 3. Results

A 35-year-old nulliparous woman presented with a fast-growing uterus due to multiple leiomyomata, filling the entire and complete abdomen. Desiring pregnancy, in previous consultation at another center, the patient had refused hysterectomy, laparotomy, or non-contained electronic power morcellation, in fear of sarcoma tumor spread. The complete surgical procedure included multiple leiomyoma enucleation, contained in-bag electronic power morcellation, uterine reconstruction, as well as, the application of an adhesion prophylactic medical product (4DryField PH™; PlantTec GmBH, Lüneburg, Germany). Postsurgical peritoneal-washing cytology showed no tissue spillage and the bags showed any puncture at the post-usage insufflation and submersible test.

The patient’s intraoperative and postoperative course was uneventful and without complication. The postoperative pain was adequately controlled with diclofenac and paracetamol parenteral every eight hours until discharge from the hospital, continuing with the same therapy at home, orally, until the fifth postoperative day. The intraperitoneal drain was removed on the first postoperative day. Due to normal food intake and bowel function, the patient was discharged on the second postoperative day. The patient is a medical professional and returned for duty after 14 postoperative days.

The total removed specimen weight was 4781 g and the histopathological examination showed an atypical or bizarre leiomyoma, exhibiting foci of cellular atypia without evidence of tumor cell necrosis or malignancy. These abnormal cell collections are benign, but are risk for cancer, therefore close-follow up by means of MRI with contrast was suggested by the oncological board to be performed after three to six months; if normal, further ultrasound controls were recommended every six months over two years.

## 4. Discussion

Laparoscopic myomectomy could be a surgical challenge, especially when surgeons are removing big surgical specimens, which produce a larger amount of debris, requiring a time-consuming clearance after non-contained morcellation. Contained morcellation has given the opportunity for surgeons to increase the level of surgical standards in terms of safety, concerning the spillage of tissue, which can occur during and after the tumor excision [20,21]. Moreover, it is possible to perform the vaginal extraction of the specimen in a containment system, even without morcellation [22]. According to the case here described and to the larger in-bag morcellation study published by us until now [15], we observed effective and safe implementation of in-bag morcellation in all cases, irrespective of patient weight or specimen size. Neither the thinnest (44.0 Kg) nor the heaviest woman (127.6 Kg) presented complications during surgery or bag manipulation and it was possible to remove big surgical specimens up to 2805 g (mean 435.5 g) without technical problems.

Besides a careful examination of the patient, other factors are important to consider when opting for the minimally-invasive approach to treat big uterine tumors, like surgeon- and institute experience, equipment availability and related costs. Otherwise, surgeons would choose an open procedure for big fibroids. With this video-article, we show that laparoscopic in-bag morcellation is a feasible option even in cases of giant fibroids. We encourage endoscopic surgeons to continue the proper learning curve to perform safely this type of procedure.

It is a known fact that morcellation of an undetected sarcoma can worsen the prognosis of the patient. In 2014 the FDA reported that the incidence of unsuspected uterine sarcoma is 1:352 for any type of sarcoma and 1:498 for leiomyosarcoma [1]. Thereafter, several studies report different incidences after hysterectomy, 0.056% (1:1784), 0.14% (1:700), 0.49% (1 in 204) [19]. The overall pooled risk was reported to be higher 0.15% (1 in 650) after hysterectomy than after myomectomy, 0.08% (1:1306) [19]. The risk appears to be age related with one study demonstrating a lower risk in patients under 45 years of age [2,3,4]. Other long-term implications like parasitic adenomyosis or leiomyomatosis are infrequently reported. Currently, there are no available methods to preoperatively diagnose a potential malignancy in leiomyoma [23,24]. Therefore, a cautious manipulation of the tumor, bags and instruments minimizes the risk of intraoperative spillage, especially in cases of larger myomata. Moreover, there is a lack of evidence regarding how much of a bag puncture would provoke tissue spillage, as well as the meaning of micro-spillage occurring in inadvertent breach of bag integrity [25]. In the presented case no tissue spillage or bag puncture occurred.

Irrespective of the low-risk of malignancy, careful presurgical image-based assessment, bag examination and the histological examination of all fibroids and peritoneal fluid samples (taken at surgery initiation and at end) should be the standard. The follow-up schedule should be individualized according to the histologic findings; re-examination is mandatory in case of suspected myoma recurrence or abnormal appearance of the peritoneal cavity at ultrasound.

On the other hand, and considering the size of this patient’s tumors and the fact that myomectomy is associated with profuse capillary bleeding and postoperative adhesions, we chose a barrier with dual properties of hemostasis and adhesion prophylaxis that can be applied evenly on all the serosal scars. The application of an adhesion prophylaxis agent is a strategy that could significantly optimize the surgical outcomes after myomectomy [26,27].

## 5. Conclusions

Contained in-bag electronic power morcellation, in the hands of an experienced and technically proficient minimally-invasive gynecologic surgeon, enables a purely laparoscopic approach for the treatment of a gigantic leiomyoma mass without enhanced risk of spillage compared to laparotomy. Postoperative complications by subsequent adhesion formation, with possible reduction of quality of life, fertility or even leading to bowel obstruction, can be reduced by intra-laparoscopic application of adhesion prophylactic agents.

## Figures and Tables

**Figure 1 jcm-11-04531-f001:**
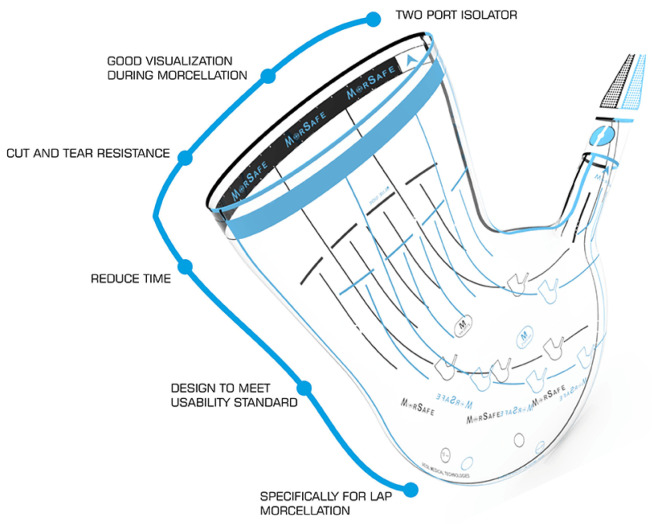
Schematic representation of the in-bag morcellation system MorSafe^®^ (reproduced with permission from Veol Medical Technologies Pvt. Ltd., Navi Mumbai, India).

**Figure 2 jcm-11-04531-f002:**
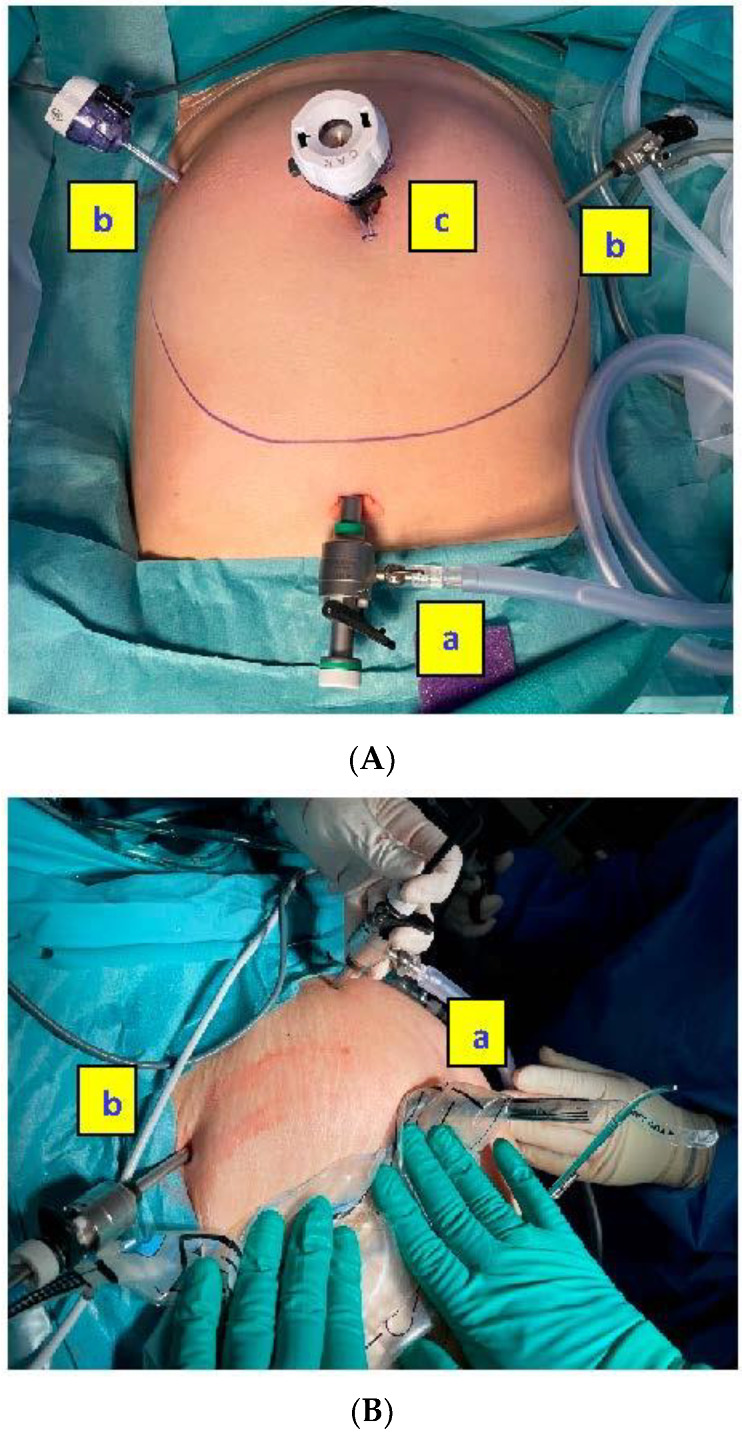
Main steps of the in-bag morcellation process (MorSafe^®^ system). (**A**) Typical port configuration to access a giant uterus consisting of (a) Telescopic port, (b) ports for surgical instruments, and (c) larger port for instruments and morcellator. (**B**) Bag insertion through the larger port (a) under visual guidance from telescopic port (b). (**C**) Morcellator (a) and telescope (b) are placed within the respective sleeves. (**D**) The morcellator is manipulated through the morcellation sleeve (a) and the camera is inserted through the telescopic sleeve (b), allowing a visual control of the morcellation process. The bag is insufflated with CO_2_. (**E**). Bag is drawn-out through the morcellator port (a) after deflation and tying of the telescopic sleeve (b) to prevent spillage. (**F**) The bag is inspected for any obvious disruption. (a) Morcellator and specimen sleeve opening, (b) Tied telescopic sleeve. (**G**) Finally, submersible puncture test is performed on the bag.

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
