# Peer review of "To Laparoscopically Preserve Fertility in Intraabdominal Giant Myoma with Application of Contained In-Bag Morcellation: Mission Impossible?"

_jcm, 2022, doi:10.3390/jcm11154531_

Round 1

Reviewer 1 Report

Dear Authors,

You have done a very good job revising the document and have adequately addressed all concerns. I believe your work will help other surgeons and I am impressed by your surgical approach and technique. 

Reviewer 2 Report

Thank you to the authors for their reply. I have no more objections.

This manuscript is a resubmission of an earlier submission. The following is a list of the peer review reports and author responses from that submission.

Round 1

Reviewer 1 Report

To Laparoscopically preserve fertility in intraabdominal giant myoma with application of contained in-bag morcellation: Mission impossible?

JCM-1709139

Thank you for this video article. This is a very important clinical topic and the authors did an impressive job performing this large myomectomy with in-bag morcellation. While the technique of minimally invasive myomectomy with in-bag morcellation is not novel, the technique used and the size of this myoma warrants publication.  Overall this document is well written and adds to the medical literature. Thank you for sharing this with us.

A few suggestions regarding this video article:

1)    Can you share the port placement used and approach on trocar placement when the fibroid took up so much space? It would be great to have a clip of the beginning of the surgery.

2)    Introduction:  

a.     I would suggest mentioning the incidence of leiomyosarcoma (FDA vs others)

b.     The authors mentioned “careful patient selection”; can they elaborate as to which patients would be ideal for this approach vs other approaches?

3)    Materials and Methods:

a.     How large was the uterus pre-op (week size or cm?) and what was the number and location of the fibroids?

b.     What type of fibroids were resected (FIGO) and was the uterine cavity entered?

c.     Which techniques were used to control bleeding (Pitressin?, compression of the uterine arteries?) and what was the blood loss for the case? Was transfusion or cell-saver used?

d.     What type of closure (and suture) was used to reconstruct the uterus and serosa post myomectomy?

4)    Discussion

a.     A brief mention of the postoperative course (pain…) and follow-up plan for this patient may be of benefit to the readers.

b.     Most surgeons would choose an open procedure for this size fibroid. Can you address how the techniques demonstrated here can be brought to other hospitals/communities as this would have a large impact to healthcare provision?

c. How was this method of adhesion prevention  selected? 

Reviewer 2 Report

The authors described an important topic concerning the problem of morcellation and possible complications in the case of, for example, sarcoma. 

The content of the work and the video material indicates that this is a case report. The authors should describe the issue in more detail in the introduction and discuss in the Discussion, because it is too short and does not exhaust the topic.